# Next-Generation Sequencing (NGS) and Third-Generation Sequencing (TGS) for the Diagnosis of Thalassemia

**DOI:** 10.3390/diagnostics13030373

**Published:** 2023-01-19

**Authors:** Syahzuwan Hassan, Rosnah Bahar, Muhammad Farid Johan, Ezzeddin Kamil Mohamed Hashim, Wan Zaidah Abdullah, Ezalia Esa, Faidatul Syazlin Abdul Hamid, Zefarina Zulkafli

**Affiliations:** 1Department of Hematology, School of Medical Sciences, Health Campus, Universiti Sains Malaysia, Kubang Kerian 16150, Malaysia; 2Institute for Medical Research, Shah Alam 40170, Malaysia; 3School of Health Sciences, Universiti Sains Malaysia, Kubang Kerian 16150, Malaysia

**Keywords:** PCR, thalassemia, sequencing, NGS, TGS, CNV

## Abstract

Thalassemia is one of the most heterogeneous diseases, with more than a thousand mutation types recorded worldwide. Molecular diagnosis of thalassemia by conventional PCR-based DNA analysis is time- and resource-consuming owing to the phenotype variability, disease complexity, and molecular diagnostic test limitations. Moreover, genetic counseling must be backed-up by an extensive diagnosis of the thalassemia-causing phenotype and the possible genetic modifiers. Data coming from advanced molecular techniques such as targeted sequencing by next-generation sequencing (NGS) and third-generation sequencing (TGS) are more appropriate and valuable for DNA analysis of thalassemia. While NGS is superior at variant calling to TGS thanks to its lower error rates, the longer reads nature of the TGS permits haplotype-phasing that is superior for variant discovery on the homologous genes and CNV calling. The emergence of many cutting-edge machine learning-based bioinformatics tools has improved the accuracy of variant and CNV calling. Constant improvement of these sequencing and bioinformatics will enable precise thalassemia detections, especially for the CNV and the homologous *HBA* and *HBG* genes. In conclusion, laboratory transiting from conventional DNA analysis to NGS or TGS and following the guidelines towards a single assay will contribute to a better diagnostics approach of thalassemia.

## 1. Introduction

The earliest clinical description of thalassemia was by the Detroit pediatricians Thomas B. Cooley and Pearl Lee, who defined a severe form of anemia among children with splenomegaly and bone deformation [1], which was later termed Cooley’s anemia. Whipple and Bradford coined the term “thalassemia” in 1936 when they described Cooley’s anemia with erythroblastic anemia [2]. The name originated from the Greek words meaning “sea” and “blood”, as the anemic patients were all of Mediterranean origin. Later, the disease was found widely throughout the Indian subcontinent, SEA, and the Middle East.

Thalassemia and hemoglobinopathies are genetic disorders caused by gene defects that hinder the normal production of hemoglobin, the main protein found in RBCs responsible for binding and carrying oxygen-RBCs from the lungs to the various body tissues. Normal adult hemoglobin consists of two pairs of α and β-chains, respectively. Synthesis of these proteins is coordinated to ensure equal production levels in erythropoietic cells [3]. When one or both copies fail to produce normal β-globin, the α-globin gene continues its normal α-globin production. The consequence of these mutations is an imbalance of α/β-globin chain synthesis, evidently in the homozygous forms, leading to the accumulation of free α-globin chains forming highly toxic aggregates [4].

Both alpha (α)- and beta (β)-thalassemia phenotypes are classified separately. α-thalassemia is classified into four phenotypes: the silent carrier, trait, HbH, and Hb Bart’s. Meanwhile, β-thalassemia is classified into four phenotypes: silent carrier (β^++^), carrier, intermedia, and major. β-thalassemia minor or trait refers to carriers of a β-thalassemia mutation with a normal β-globin on the other allele. They are clinically asymptomatic. β-thalassemia major is characterized by infancy-onset severe anemia that requires lifelong blood transfusion for survival [5]. In β-thalassemia major, the excess unpaired α-globin chains aggregate to form inclusion bodies that damage RBC membranes, causing intravascular hemolysis, damage and apoptosis of erythroid precursors, and ineffective erythropoiesis [6]. β-thalassemia intermedia is defined as having an intermediate thalassemia condition between minor and major [7]. Most patients with β-thalassemia intermedia are homozygotes or compound heterozygotes for β-thalassemia [8]. Rarely, a simple carrier of β-thalassemia can be symptomatic, such as in the case of co-inheritance of segmental duplication of the α-chain, which increases the imbalance ratio of β- and α-tetramer, resulting in a more severe phenotype [9,10].

The separate phenotype classification of α- and β-thalassemia can be perplexing, especially for patients with a complex genotype such as concomitant α- and β-thalassemia. Thus, new symptomatic thalassemia called transfusion-dependent thalassemia (TDT) and non-transfusion-dependent thalassemia (NTDT) were proposed primarily to classify various α- and β-thalassemia and variants [11]. TDT refers to patients requiring lifelong regular blood transfusions to survive [12]. NTDT is a term to describe patients that do not require such lifelong regular transfusions for survival, although they may require occasional or even frequent transfusions in certain clinical settings and for defined periods [13]. NTDT encompasses five clinically distinct forms: β-thalassemia intermedia, hemoglobin E/β-thalassemia (mild and moderate forms), α-thalassemia intermedia (Hb H disease), hemoglobin S/β-thalassemia, and hemoglobin C thalassemia [11].

Mutations in the *HBA* and the *HBB* are associated with α- and β-thalassemia or variants, respectively. They are extremely heterogeneous, with more than a thousand types of mutations comprising single nucleotide variation (SNV), indels, segmental deletions and duplications, and segmental inversion [14,15,16]. The β-thalassemia is mainly caused by SNV, while segmental deletions are the common cause of α-thalassemia. Deletions of the *HBB* cluster lead to either high persistence of fetal hemoglobin (HPFH), delta beta (δβ-thalassemia), or β-thalassemia and rarely are concomitant with segmental inversions and donor insertion. Segmental duplications are more common in the α-globin cluster than in the β-globin cluster. The importance of molecular diagnosis in thalassemia prevention and its utilization in the clinical decision has become a standard practice in patient management. The variable phenotypic expression renders molecular study crucial for the confirmation of diagnosis, treatment, and counseling purposes.

## 2. Conventional DNA Analysis

Before the invention of PCR, the diagnosis of β-thalassemia was performed by linkage analysis using restriction fragment length polymorphism (RFLP) [17,18], and Southern transfer and hybridization or Southern-blot analysis [19,20]. Although the enzymatic amplification of DNA was introduced in 1971 [21], the clinical application was described years later for prenatal diagnosis of sickle cell anemia. Using less than 100 times of DNA or as little as 20 ng of DNA, improved sensitivity of DNA hybridization to Phosphorus-32 (^32^P) end-labeled oligonucleotide probe (isotope) was achieved [22]. The enzymatic amplification of DNA was later termed polymerase chain reaction (PCR) [23]. This enabled the use of a non-radioactive probe of horseradish peroxidase-labeled oligonucleotides for the dot-blot analysis [24,25].

### 2.1. Reverse Dot-Blot Analysis

Later, the dot-blot analysis was reversed. The horseradish peroxidase-labeled sequence-specific oligonucleotide probes were spotted onto the nylon membrane, allowing simultaneous hybridization reaction of an entire series of sequences [26]. Customized reverse dot blot has been used in many populations [27,28,29,30,31,32,33,34]. Typically, a reverse dot-blot hybridization requires three steps; immobilization of allele-specific oligonucleotide probe to a nylon membrane, amplification of the targeted region of DNA using a biotinylated primer, and hybridization of the biotinylated DNA to the probe-bound nylon membrane using streptavidin-alkaline phosphatase and color substrates.

### 2.2. Gap-PCR

Gap-PCR is the amplification of the excess segment of deletional thalassemia types. By amplifying the excess segment, an estimation of deletional types is made based on the size of the amplicons. The most commonly used gap-PCR is the multiplex gap-PCR detecting common deletional α-thalassemia [35] and ααα^anti 3.7^ and ααα^anti 4.2^ triplications [36]. Other gap-PCR useful to detect deletional forms of α- and β-thalassemia include the gap-PCR for the HPFH, δβ-thalassemia, and β-thalassemia [37,38,39].

### 2.3. Amplification Refractory Mutation System (ARMS) or Allele-Specific Polymerase Chain Reaction (ASPCR)

Wu and his colleagues proposed a simple allele-specific oligonucleotide PCR (ASPCR) approach that did not require enzyme digestion and blot hybridization for DNA analysis of sickle cell anemia [40]. Later, the same concept dubbed amplification refractory mutation system (ARMS) was described [41]. It was based on allele-specific primers with a modified 3′ end to exactly match the point mutation of choice and the inclusion of a mismatch at the fourth nucleotide of the 3′ end to deliver the extra specificity. The lack of 3′ to 5′ exonuclease activity of Taq polymerase reduced its ability to extend efficiently. These primers can amplify SNV and small indel mutations, whereas homozygosity or heterozygosity is detectable using oligonucleotide primers that perfectly match the wild-type sequences at the same position of the same mutation. Typically, an ARMS requires the amplification of DNA fragments by allele-specific PCR followed by gel electrophoresis of the amplified DNA. Customized multiplex ARMS have also been used in many populations [33,41,42,43,44,45,46,47,48,49,50,51,52].

### 2.4. Sanger Sequencing

In 1975, a new technique allowing the determination of sequence in a specific region of the DNA chain was described [53]. Later, the isotope labeling was replaced by fluorophore-labeled dNTPs that were covalently attached to the oligonucleotide primer used in enzymatic DNA sequence analysis [54]. Then, the autoradiography step was replaced by a computer that acquires the sequence information directly [55]. With the introduction of PCR, sequencing of specific regions enabled the detection of rare β-thalassemia [56,57]. In 1990, the separation of fluorescently labeled DNA fragments by capillary electrophoresis (CE) was introduced [58]. Sequencing was used as a subsequent method following targeted PCR, RFLP, and dot-blot analyses [29,33,50,51,59], and became the gold-standard for diagnosis of thalassemia mutations and copy number variation (CNV) breakpoint analysis.

### 2.5. Multiplex Ligation Probe-Dependent Analysis

Ligation-dependent PCR was introduced for the detection of the Hepatitis C virus [60]. In the same year, a ligation-dependent PCR was patented as multiplex ligations-dependent amplification (MLPA) [61]. Later, MLPA detection of multiple diseases was presented [62] and made commercially available for hereditary disorders, tumor profiling, and methylation status in diagnostic and research by the MRC Holland (Amsterdam, The Netherlands). Each MLPA probe set comprises two probes. The first half contains target-match attached probes with a universal primer. The other half has an extra stuffer sequence meant to produce uniquely sized amplicon upon hybridization, ligation, and amplification. Upon hybridization to the target DNA, ligation of both probes permits measurable amplification of the probes. For SNP detection, the 3′ mutation-specific MLPA probe produced a quantifiable signal when annealed to the target sequence. Using the MLPA probe mix (MRC-Holland, Amsterdam, The Netherlands), α-thalassemia [63,64], large segmental duplications among symptomatic β-thalassemia carriers [65,66,67,68,69], *HBG1-HBG2* deletion [64], and εγδβ-thalassemia [70,71] were discovered.

## 3. Advanced Molecular Techniques towards the Single-Assay DNA Analysis

For many years, DNA analyses of Mendelian disease such as thalassemia have relied on two major steps: identifying common mutations via various targeted DNA analysis methods, and rare mutation discovery via Sanger sequencing and MLPA analysis. An extensive DNA analysis detects *HBA*, *HBB*, and *HBD* variants and CNV via Sanger sequencing and MLPA, respectively (Figure 1). Indels are common in *HBB*, thus extra reads are required to accommodate the frameshifted heterozygous mixed Sanger sequencing reads. Homozygous variants must consider the possibility of accompanying deletion on the other allele, so an additional test to rule it out is a must.

For the detection of deletional thalassemia, amplification of the *HBA2* gene [35] and cluster-spanning amplicon were used [37,38,39] to represent the normality of the other allele, respectively. Homozygous deletional detected by these Gap-PCR must be accompanied by MLPA to rule out rare CNV. MLPA uses a comparative sample to known reference analysis and spanning probe patterns for the identification of CNV. In most cases, the interpretation is straightforward, but cannot be used as a standalone technique because of its limited probes. For deletions and duplications spanning across the *HBA* and *HBB* clusters, validations by long-range PCR are possible because of the many probes available within these clusters. However, larger deletions and duplications breakpoints are harder to estimate owing to the limited number of the MLPA probes available beyond *HBA* and *HBB* clusters.

### 3.1. Next-Generation Sequencing (NGS)

Two years after the completion of the Human Genome Project (HGP), new sequencing technology emerged. Next-generation sequencing (NGS) enabled scalable high throughput sequencing by adapting parallel detection of small DNA fragments (150–1000 bp). Complete sequencing can be achieved by whole genome sequencing (WGS), while enrichment enables exome, targeted, RNA, and methylation sequencing. NGS chemistry differs between platforms. For instance, Illumina uses clonal array formation and a proprietary reversible terminator for large-scale sequencing. Adaptor and index ligated DNA fragments are immobilized on flowcell, where each fragmented DNA is isothermally amplified, generating millions of DNA clusters. The clusters are excited by light source and the characteristic fluorescent signal is emitted (sequence by synthesis, SBS). The number of cycles determines the length of the reads. The emission wavelength and the signal intensity determine the basecalls (Figure 2).

Prior to variant calling, reads are mapped against the reference sequence to identify the region of origin for each sequencing read. Among the most widely used mappers are the Genome Analysis Toolkit (GATK) [72], preferred Burrow-Wheeler Aligner-Maximal exact match (BWA-MEM) [73], Bowtie2 [74], minimap2 [75], and Scalable Nucleotide Alignment Program (SNAP) [76]. SNAP conveniently compresses, sorts, mark-duplicates, and indexes the final output. Various highly accurate variant and indel calling tools employ different approaches and outputs. GATK [72], FreeBayes [77], and SAMtools [78] rely on bayesian approaches. DeepVariant utilized deep neural networks [79], while Strelka2 [80] used a novel mixture-model-based estimation to call variants and indels.

Unlike variants and indel genotyping, detection of large rearrangements in copy-number variants (CNV) from NGS data is challenging owing to the technology’s natural limitations such as short read lengths and GC-content bias [81]; however, options are aplenty. Control-FREEC uses a LASSO-based algorithm [82]; DELLY uses short and long-range paired-end mapping and split-read analysis [83]; and CNVkit uses in-target and off-target regions, bias correction using rolling median, and circular binary segmentation (CBS) [84] to call CNV. Tools dedicated to analyzing CNV from exome sequencing reads include ExomeDepth, which uses a beta-binomial model to generate a likelihood value, the hidden Markov model (HMM) to combine the likelihood across multiple exons, and maximum likelihood Viterbi algorithm to provide a set of calls for each sample [85]; CoNIFER uses combined read-depth with singular value decomposition (SVD) normalization and ±1.5 SVD-transformed standardized z-scores reads per thousand bases per million read sequenced (ZRPKM) (copy number inference from exome reads) [86]; and FishingCNV uses principal component analysis (PCA) of the RPKM, CBS test sample, and comparing segment coverage against the control set distribution [87]. FishingCNV is also available in a graphical software package. The CNV tools and their features are summarized in Table 1.

User-friendly interface genotyping CNV is also available. Detection and Annotation of Copy Number Variations (DeAnnCNV) is an online detection and annotation of CNVs from exome sequencing data that extracts the shared CNVs among multiple samples and provides annotations for the detected CNVs and associated genes [88]. CovCopCan uses normalized read count value (NRC) for each amplicon to generate a CNV detection algorithm based on z-score. False positive is minimized by applying a two-stage ratio to the amplicons, while false negative of two CNV per chromosome is countered by merging the CNV area. A two-dimensional CUSUM chart, local regression curve, and variant call format (VCF) file can be generated [89].

Another hurdle of the CNV analysis is the interpretation of the text format output that lacks genetic or clinical data annotations, so users need to compare all the positions of CNV and use any annotation tools to determine their genetic meaning [90]. Several tools dedicated to interactive and dynamic visualization of CNV have been developed. Scripps Genome Annotation and Distributed Variant Interpretation Server (SG-ADVISER) is a suite consisting of an annotation pipeline and a web server that provides known and predicted information about genetic variants by performing in-depth annotations and functional predictions for variants and CNVs [91]. A web-based application, inCNV, integrates and prioritizes CNV-tool results with user-friendly interfaces and analyzes the importance of called CNVs by generating CNV annotations from Ensembl, Database of Genomic Variants (DGV), ClinVar, and Online Mendelian Inheritance in Man (OMIM) [90]. reconCNV uses delimited result files from most NGS CNV callers to produce an interactive dashboard that can be visualized as Hyper Text Markup Language (HTML) output [92]. Meanwhile, a web server CNVxplorer provides a functional assessment of CNVs in a clinical diagnostic setting by mining a comprehensive set of clinical, genomic, and epigenomic features associated with CNVs [93]. In R, called copy number and beta allele frequency (BAF) data can be visualized by KaryoploteR [94], Gviz [95], and CopyNumberPlots [96].

Targeted sequencing (TS) is the most economical approach for thalassemia as the clusters are small. Ideally, a TS should be able to detect point mutations in *HBA*, *HBB*, *HBD*, and *HBG* and include uniform reads covering the adjacent genes for rare CNV discovery and breakpoint estimation. Adding CNV spanning reads allows accurate direct detection of common CNV. TS has been described to be superior to the conventional DNA analysis strategy. Using the GATK variant calling pipeline coupled with an in-house CNV tool, simultaneous genotyping of globin genes and genetic modifiers had better performance than traditional routine screening [97]. TS of the *HBB* gene and common deletional α-thalassemia of -α^3.7^ and -α^4.2^ using Ion Torrent showed genotype concordance to that of the conventional PCR [98]. A combination of gap-PCR genotyping of the common deletional α-thalassemia and variant genotyping of *HBA* and *HBB* genes by NGS produced higher sensitivity detection than using the MCV, MCH, and HbA2 screening strategy [99]. Higher sensitivity and specificity were achieved with prioritized CNV genotyping in the α-globin cluster along with *HBA* and *HBB* variants [100,101]. New *HBB* cluster deletion dubbed Inv-Del English V εγδβ-thalassemia (HbVar 2935) of 122.6 kb deletion with 56 kb and 82 bp inversion was characterized using customized targeted panel and RPKM analysis [71]. Zebisch et al., identified a novel variant of εγδβ-thalassemia using MLPA and comparative genomic hybridization (CGH) that was missed by NGS; however, CNV analysis was not mentioned [102]. Whole exome sequencing (WES)-based CNV analysis using an exome hidden Markov model (XHMM) [103] identified α-globin cluster duplication in severe β-thalassemia carriers [68].

Short reads mapping of highly homologous regions such as in the *HBA1*, *HBA2*, *HBG1*, and *HBG2* genes is still challenging for NGS. A customized bioinformatics pipeline called NGS4THAL incorporates realignment of the ambiguously mapped reads derived from the hemoglobin gene cluster homologs for variants and indels calling, coupled with multiple tools for CNV discovery to improve genotyping sensitivity and specificity [104]. Using customized long-read WGS of 400 bp per read, patients with rare forms of α- and β-thalassemia were diagnosed [105]. Longer-range reads can also be achieved using link-read sequencing. It utilized multiple bar-coding of the gDNA prior to fragmentations. This allows whole-genome phasing that provides haplotype information valuable for genetic diseases. Successful haplotype-phasing for embryo selection in preimplantation genetic testing has been demonstrated using link-read sequencing for an --SEA carrier partner [106].

### 3.2. Third-Generation Sequencing (TGS)

Third-generation sequencing (TGS) employs single-molecule sequencing (SMS) that directly sequences individual DNA or RNA strands present in a sample of interest without prior clonal amplification of the DNA [107]. Uninterrupted DNA polymerase incorporates fluorescently labeled deoxyribonucleoside triphosphates (dNTPs), generating continuous DNA synthesis [108]. Pacific Bioscience (PacBio) uses single-molecule real-time (SMRT) isoform sequencing (Iso-Seq). DNA library is created by transforming double-stranded DNA with ligation adapters into circular single-stranded DNA (SMRTbell). Base-calling of the SMRTbell occurs in a chip called an SMRT cell that contains a photonic nanostructure called zero-mode waveguide (ZMW) wells. Immobilized polymerase on the surface of each well initiates DNA replication, producing an interpretable fluorescent pulse (Figure 3) [109]. The Sequel II device outputs Circular Consensus Sequence (CCS) reads. CCS features sequencing of the same molecule multiple times, generating multiple subreads of the SMRTbell library called highly accurate long reads (HiFi reads), thus improving the accuracy of SNV calling [110].

Meanwhile, in nanopore-based SMS, DNA molecules are individually translocated through nanoscale pores that only permit the passage of single-stranded DNA in a strict linear sequence [107]. Oxford Nanopore Technology (ONT) sequencers measure changes in ionic current when the DNA fragments pass through protein nanopores in a semi-synthetic insulated membrane. A single library comprises a DNA fragment, an adapter-bound motor protein, and a tethering molecule that chains the DNA to the nanopore and membrane [111,112]. Motor protein controls the translocation speed of the DNA, and it feeds DNA bases through the pore. DNA passing through the nanopore leads to a continual change in current, known as the “squiggle” stored by the MinKNOW ™ software (version 20.10, Oxford, UK). Using a neural network algorithm, MinKNOW translates the squiggle into nucleotides using graphical processing units (GPUs) in real time [113] (Figure 4).

Long-read sequencing is superior for CNV detection as a single read can span across exons, genes, pseudogenes, highly duplicated sequences, and CNV, but fell short owing to a higher indel error rate [114,115]. Various state-of-the-art correction tools are available to counter such errors and can be tested during pipeline optimization [116]. Error correction can be done before or after the genome assembly. Assemblers such as Flye [117], wtdbg2 [118], Shasta [119], and CONSENT [120] assemble the raw data using minimap2 pairwise aligner [75] before polishing/correcting the assembly. Conversely, MECAT (in-house aligner) [121], Canu [122] (using MinHash Alignment Process (MHAP) aligner [123]), Falcon [124] (using basic local alignment with successive refinement, BLASR aligner) [125], and NECAT (in-house assembly module) [126] correct error reads and then assemble them.

Improved variant calling using deep neural network (DNN) algorithms for ONT using PEPPER-Margin-DeepVariant [127], NanoCaller [128], and Clair3-trio [129] has been demonstrated. For variant calling of the HiFi reads, accurate variant calling has been shown using GATK Haplotypecaller [72,110], DeepVariant [79], and HELLO [130]. SMRT reads can be analyzed using its in-house PacBio structural variant (SV) calling and analysis tools (pbsv). Numerous CNV callers supporting both SMRT and ONT reads are available. Sniffles2 uses adaptive clustering (repeat aware), followed by a fast consensus sequence, and a coverage-adaptive filter [131], while cuteSV2 implements heuristic signature purification and a specific-designed scanning line [132] to call CNV. Interspersed duplications, tandem duplications, and insertions of novel elements can be detected by SVIM [133], while SV in low read depth WGS can be detected by the neural-network-based algorithm implemented by NanoVar [134]. Generally, both NGS and TGS workflows begin with library preparation, followed by sequencing of the prepared library, quality assessment, and reads’ trimming. However, they differ in reads’ assembly as TGS requires reads’ polishing either before or after mapping, followed by SNV, indels, and SV calling (Figure 5).

Utilizing the CCS, targeted sequencing using specific amplicons aimed at genotyping common -α^3.7^, -α^4.2^, --^SEA^, and *HBA* and *HBB* SNVs showed a complete concordance to the conventional PCR-based genotyping [135]. Later, the strategy was modified to allow genotyping of more deletional thalassemia forms. Termed a comprehensive analysis of thalassemia alleles (CATSA), the targeted SMRT sequencing was tested on 1759 samples and successfully genotyped common and rare thalassemia SNV, indels, and CNV [136]. Then, CATSA was used to genotype 100 samples, showing abnormal hematological parameters, but were uninformative during conventional genetic diagnosis by RDB and Gap-PCR (genotyping -α^3.7^, -α^4.2^, --^SEA^ only). Ten rare mutations were found [137]. Recently, Li et al., demonstrated detections of HbH disease caused by various deletional and non-deletional α-thalassemia mutations and concomitant α-thalassemia with point mutations and indel types of β-thalassemia [138]. Using MLPA and SMRT, an α-globin gene cluster 27,311 bp deletion (--^27.3^/αα), an HS-40 region 16,079 bp deletion, a rearrangement of -α^3.7^α1α2 on one allele, a *ß*-globin gene cluster *HBG1-HBG2* 4924 bp deletion, and a 15.8 kb deletion α-thalassemia were characterized [63,64].

In a non-invasive prenatal testing (NIPT), a long-range 20 kb amplicon sequenced by ONT and NGS was used to phase parental haplotypes to determine fetal inherited haplotypes by the relative haplotype dosage (RHDO) analysis, and successfully genotyped 12 of the 13 fetal thalassemia statuses [139]. Comparative analysis of ONT and Sanger sequencing showed 100% concordance of *HBB* genotyping in a small-scale study in Tanzania [140]. Liu et al., demonstrated genotyping of homozygous—SEA deletion embryos using simple read density plots across the *HBA* locus, thus showing the feasibility of ONT sequencing for preimplantation genetic testing (PGT) [141].

## 4. Discussion

Conventionally, differential diagnosis examination of hematological parameters and patient’s phenotype is used to decide the DNA analysis. It is ambiguous owing to the phenotype variability and the conventional DNA analysis test limitations. Generally, point mutations and indels are detected by ARMS-PCR or sequencing and large deletions are detected by gap-PCR or MLPA. A homozygous β-thalassemia detected by the ARMS-PCR and sequencing may not be a true homozygote, but rather a compound heterozygous with a deletional β-thalassemia or δβ-thalassemia that must be ruled out by the gap-PCR, MLPA, or cascade screening. Misdiagnosis could occur in complex genotyping, which alters the hematological parameters, such as in the mild β-thalassemia/δ-thalassemia with normal HbA_2_. The multiple methods are labor-intensive and prolong the laboratory turn-around time (TAT). Conversely, NGS and TGS permit simultaneous mutation detections of SNV, indel, and CNV, thus genotyping concomitant α- and β-thalassemia and rare variants. It improves the DNA analysis precision and promotes a better understanding of the genotype–phenotype relationship.

NGS and TGS also allow minimal DNA usage and increase throughput by sample multiplexing, hence reducing per-sample cost and TAT. Substantial costs for NGS and TGS are from the library preparation and sequencing with small amounts by the analysis. For conventional DNA analysis, the cost comes mainly from the reagents (i.e., PCR master mix) and may be lower than that of NGS and TGS. However, it can never match the resolutions of NGS and TGS. The specific challenge of NGS and TGS is the technically demanding bioinformatics analyses. While TGS permits phasing during genome assembly, haplotype-phasing is possible for short reads using pangenomic mapping. A pangenome integrates whole-genome sequences from multiple individuals to represent genetic diversity [142,143]. The reference pangenome would potentially address the biases and errors of the single linear reference (GRCh38) and is managed by the Human Pangenome Reference Consortium (HPRC) [144]. However, tools and pipelines for graph-based pangenome mapping are more complex and limited. Table 2 outlines the advantages of NGS and TGS over conventional DNA analysis.

## 5. Conclusions

NGS and TGS are useful for simultaneous SNV, indels, and SV genotyping. Owing to the mutation heterogeneities, a single-assay TS requires uniform reads, long reads for haplotype-phasing of the homologous genes (*HBA* and *HBG*), and breakpoint spanning reads for direct detection of the common deletions and duplications. The improvement of the sequencing technology to reduce the error rates and limitations, such as the upcoming Pacbio Revio long read and Onso short read systems, the Illumina Complete Long-Read technology, and state-of-the-art reads error correction tools for TGS, can be leveraged for the detection of the heterogenous thalassemia mutations.

These technologies will not replace conventional screening and PCR-based genotyping thanks to their ease of use. Complex thalassemia such as in the HKαα (Hong Kong αα) allele containing both the -α^3.7^ and ααα^anti4.2^ can be difficult to diagnose by NGS. Furthermore, conventional PCR can validate the genotyping by NGS and TGS. Traditional differential diagnosis by analyzing blood test data before DNA analysis allows genotype–phenotype correlation, thus spotting any human error during sample handling.

The key benefit of NGS and TGS over conventional DNA analysis is in their ability to genotype α- and β-thalassemia simultaneously, allowing complete diagnosis of the thalassemia and the genetic modifiers, which is crucial for genetic counseling. Besides, sequencing data banking permits re-analysis when necessary. Instruments such as the computational infrastructure for the data analysis, a skilled bioinformatics technician, properly documented TS, and the bioinformatics pipeline development and optimization that follow recommended guidelines are expensive for any start-up laboratory, but will ease as sample throughput increases. Towards clinically relevant usage of these technologies, several recommendations and guidelines have been introduced [145,146,147,148].

## Figures and Tables

**Figure 1 diagnostics-13-00373-f001:**
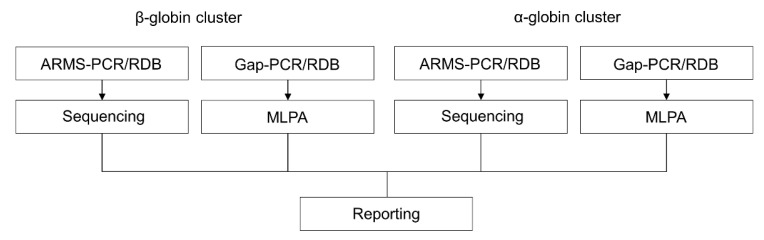
Comprehensive conventional PCR-based genotyping. ARMS-PCR and gap-PCR detect common SNV/indel and CNV, respectively, while the reverse dot-blot (RDB) can be customized to detect SNV/indel and CNV simultaneously. Sequencing and MLPA genotype unknown SNV/indel and CNV, respectively.

**Figure 2 diagnostics-13-00373-f002:**
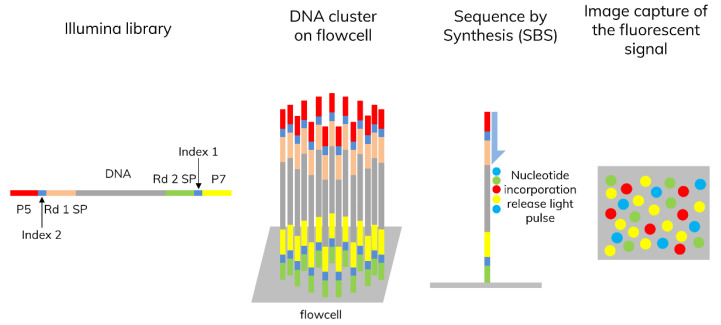
Illumina sequencing by synthesis (SBS). DNA templates are immobilized on a flowcell. When nucleotides are incorporated onto the DNA strands, they release light pulses that are captured by the sequencer and output the bases read from each cluster, along with the quality metrics for each base.

**Figure 3 diagnostics-13-00373-f003:**
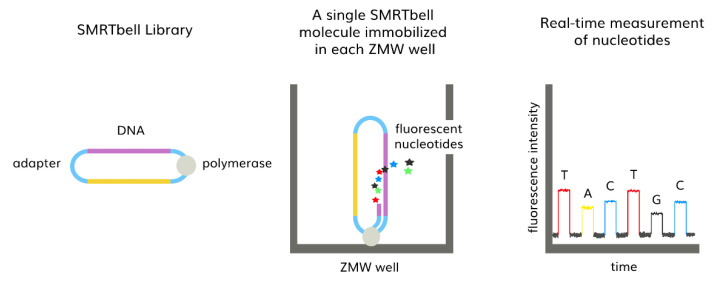
PacBio’s SMRTbell library base calling in SMRT cell containing millions of ZMW wells. Anchored polymerase incorporates the nucleotides, emitting light that is measured in real-time. A base-calling algorithm translates the light into DNA sequence.

**Figure 4 diagnostics-13-00373-f004:**
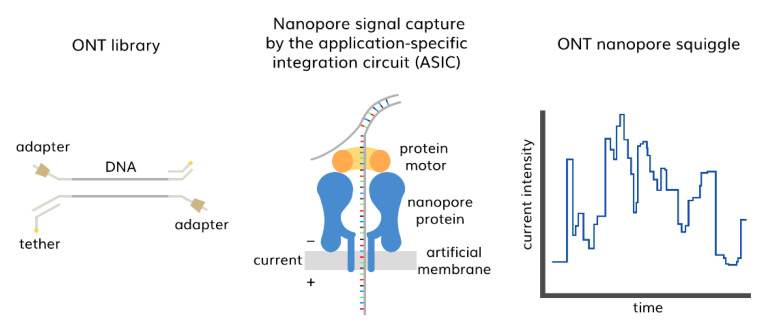
ONT’s library base calling in nanopore protein.

**Figure 5 diagnostics-13-00373-f005:**
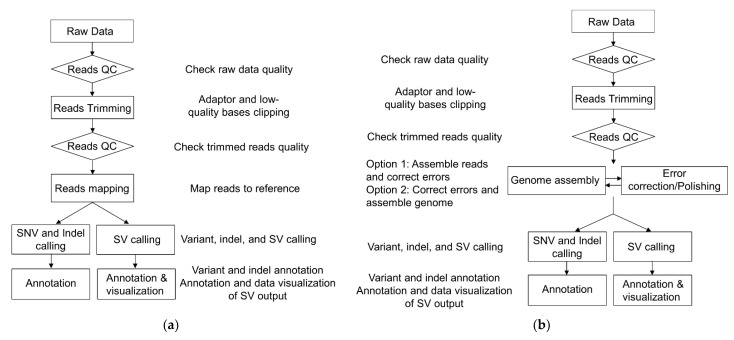
Overview of bioinformatics workflow for (**a**) NGS and (**b**) TGS, requiring additional error correction/polishing, either by assembling the genome first followed by error correction or vice versa. Both pipelines use binary alignment map (bam) for SNV, indel, and SV calling. SNV and indel are called simultaneously, while SV calling uses specialized tools/software.

**Table 1 diagnostics-13-00373-t001:** CNV tools and their features.

Tool	Algorithm	Highlight
Control-FREEC	LASSO-based, Gaussian mixture models (GMM)	Output BAF from SAM pileup or ratio and copy number calls of each segment
		Use GC content and mappability profiles to normalize read count if control sample is unavailable
DELLY2	Graph-based paired-end clustering and k-mer filtering for split-read analysis	Call SV from distinct insert sizes PE libraries
		Output VCF containing SV quality prediction
		Support short and long reads
CNVkit	Circular Binary Segmentation (CBS), HaarSeg, HMM	Primarily for hybrid capture sequencing
		Use on- and off-target reads to call CNV
		Support amplicon sequencing-based TS
		Multiple segmentation algorithms to choose from
ExomeDepth	Beta-binomial model, HMM, maximum likelihood Viterbi algorithm	An R package works on Windows and UNIX systems
		Source read count data from multiple samples to build optimized reference sets
CoNIFER	Singular value decomposition and z-scores reads per thousand bases per million read sequenced (SVD-ZRPKM)	Use Matplotlib and Pyplot to generate arbitrary segment of the SVD-ZRPKM data
		Calculate batch effect biases by concurrently analyzing multiple samples suitable for large sample sets
FishingCNV	PCA	Support CLI and GUI for Windows and UNIX systems
		Compare coverage depth in test samples and use PCA to remove batch effect

**Table 2 diagnostics-13-00373-t002:** Comparison of DNA analysis of thalassemia using the conventional method, NGS, and TGS.

Feature	Conventional	NGS	TGS
DNA usage	High	Low	Low
Mutation detection	Method-dependent	Simultaneous	Simultaneous
Haplotype-phasing	Not relevant	Yes ^1^	Yes
TAT	Long	Short	Short
Per sample cost	Variable	Uniform	Uniform
Technical difficulty	Low	High	High

^1^ Via pangenome mapping.

## Data Availability

The data presented in this study are available upon request from the corresponding author.

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
