# Peer review of "Next-Generation Sequencing (NGS) and Third-Generation Sequencing (TGS) for the Diagnosis of Thalassemia"

_diagnostics, 2023, doi:10.3390/diagnostics13030373_

Round 1
Reviewer 1 Report
Dear Authors
I reviewed the your manusctipt titled “Next-Generation Sequencing (NGS) and Third-Generation 2 Sequencing (TGS) for the Diagnosis of Thalassemia” An excellent review article written in this field. I would like to congratulate to all of you.
I liked the article very much, except for one comment. My comment is that there are four clinical types in beta thalassemia like alpha thalassemia, silent carrier(B++), carrier, intermedia and major.
Author Response
Reviewer 1
My comment is that there are four clinical types in beta thalassemia like alpha thalassemia, silent carrier(B++), carrier, intermedia and major.
Reply
Thanks for the comment. Correction done. To refer pg 2.
Reviewer 2 Report
This review aims to describe how Next-Generation Sequencing (NGS) and Third-Generation Sequencing (TGS) are more appropriate and valuable for DNA analysis for the Diagnosis of Thalassemia.
1. It is important to have a section to explain how different biological pipelines are used to analysis and curated the SNVs and CNVs calling. However, they are too technical and lack of clarity. For example "Control-FREEC uses a LASSO-based algorithm [82], DELLY uses short and 213 long-range paired-end mapping and split-read analysis [83], CNVkit uses in-target and 214 off-target regions, bias correction using rolling median, and circular binary segmentation 215 (CBS) [84] to call CNV" I suggest a table listed the name of algorithm used and purpose of analysis and its application in genotyping/CNV call and necessary requirement special to alpha and beta Thalassemia will be helpful .
2. Not sure what is the purpose of the last section. This part discussing TGS application in non-invasive prenatal testing (NIPT) and PGT-M. Since other sections are describing findings based on technology and platform related to screening and diagnosis of Thalassemia, it is hard to understand the logic here and also lack of good content.
3. To make this review with addon value to this field, shorten the NGS and TGS method description but add in a table summarize how NGS and TGS has advantage over convention on DNA materials required, mutation detection, phasing analysis, findings of rare and concomitant α- and beta thalassemia, TAT and cost will be more meaningful for reference and discussion.
Author Response
Reviewer 2
This review aims to describe how Next-Generation Sequencing (NGS) and Third-Generation Sequencing (TGS) are more appropriate and valuable for DNA analysis for the Diagnosis of Thalassemia.
- It is important to have a section to explain how different biological pipelines are used to analysis and curated the SNVs and CNVs calling. However, they are too technical and lack of clarity. For example "Control-FREEC uses a LASSO-based algorithm [82], DELLY uses short and 213 long-range paired-end mapping and split-read analysis [83], CNVkit uses in-target and 214 off-target regions, bias correction using rolling median, and circular binary segmentation 215 (CBS) [84] to call CNV" I suggest a table listed the name of algorithm used and purpose of analysis and its application in genotyping/CNV call and necessary requirement special to alpha and beta Thalassemia will be helpful.
Reply
- Thanks for the comment. Correction done. Have added table 1- listing the CNV tools, their algorithms and features. To refer pg 6.
- Not sure what is the purpose of the last section. This part discussing TGS application in non-invasive prenatal testing (NIPT) and PGT-M. Since other sections are describing findings based on technology and platform related to screening and diagnosis of Thalassemia, it is hard to understand the logic here and also lack of good content.
Reply
- Thanks for the comment. Correction done. Rephrase the last paragraph of the TGS section on NIPT. To refer pg 10.
- To make this review with addon value to this field, shorten the NGS and TGS method description but add in a table summarize how NGS and TGS has advantage over convention on DNA materials required, mutation detection, phasing analysis, findings of rare and concomitant α- and beta thalassemia, TAT and cost will be more meaningful for reference and discussion.
Reply
3. Thanks for the comment. Correction done. Add information in discussion focussing on comparison between conventional DNA analysis, NGS and TGS and also adding table 2 -summarizing the comparison. To refer pg 10.